# "Betaone" barley water extract suppresses ovariectomy-induced osteoporosis *in vivo* and RANKL-induced osteoclast differentiation *in vitro*

**Yongjin Lee[1], Hyun-Jin Lee[2], Kwang-Jin Kim[3], Han-Byeol Shin[3], Yoon-A Shin[3], Holim Jin[3], Ju Ri Ham[4], Soo-Young Choi[5], Mi-Ja Lee[6]\*, Mi-Kyung Lee[5]\*, Young-Jin Son[3]\***

**1** Department of Nutritional Science & Food Management, Ewha Womans University, Seodaemun-gu, Seoul, Republic of Korea, **2** The DABOM Inc, Seodaemun-gu, Seoul, Republic of Korea, **3** Department of Pharmacy, Sunchon National University, Suncheon-si, Jeollanam-do, Republic of Korea, **4** Mokpo Marin Food-Industry Research Center, Mokpo-si, Jeollanam-do, Republic of Korea, **5** Department of Food and Nutrition, Sunchon National University, Suncheon-si, Jeollanam-do, Republic of Korea, **6** Crop Foundation Research Division, National Institute of Crop Science, Rural Development Administration (RDA), Wanju-si, Jeollabuk-do, Republic of Korea

\* esilvia@korea.kr (M-JL); leemk@scnu.ac.kr (M-KL); sony@sunchon.ac.kr (Y-JS)

## Abstract

Betaone is a variety of barley developed by the Korea Rural Development Administration. This study investigated the anti-osteoporosis effects of Betaone barley water extract (B1W) on ovariectomy (OVX)-induced bone loss in mice. To elucidate its mechanism, the effect of B1W on osteoclasts was assessed by measuring the protein expression of nuclear factor-activated T cells c1 (NFATc1), the expression of genes involved in osteoclast differentiation, and bone pit assays. B1W (300 mg/kg/day) significantly increased bone mineral density and bone volume fraction, but decreased trabecular separation compared to the OVX group. B1W also showed a trend towards decreasing serum C-telopeptide of collagen type 1 levels in OVX mice. Additionally, B1W reduced the expression of NFATc1 and downregulated the mRNA expression levels of various marker genes such as c-Fos, tartrate-resistant acid phosphatase (TRAP), cathepsin K (CTSK), dendritic cell-specific transmembrane protein (DC-STAMP), and osteoclast-associated Ig-like receptor (OSCAR). B1W reduced the osteoclast activity in the receptor activator of nuclear factor-κB ligand (RANKL)-treated osteoclasts by inhibiting the mitogen-activated protein kinase (MAPK) pathway. Based on the results, B1W can be considered a useful candidate for a therapeutic agent for treating conditions of bone loss and could also be used as an ingredient in health supplements.

## Introduction

As the elderly population continues to grow, there has been a rapid rise in the incidence of age-related diseases such as osteoporosis [1]. In osteoporosis, there is a decrease in bone mineral density including calcium content and a deterioration of the microstructures, making

**Data availability statement:** All relevant data are within the manuscript and its Supporting information files.

**Funding:** This research was funded by the Cooperative Research Program for Agriculture Science and Technology Development, South Korea (Grant No. PJ013524022022 and IRIS Project No. RS-2023-00224188).

**Competing interests:** The authors declare that they have no conflict of interest.

the bone less dense [2]. In addition, the expression of various proteins within the bone is also altered [3]. These changes increase the structural vulnerability of bones, significantly raising the susceptibility to fractures [4]. Osteoporosis is a result of an imbalance between bone formation and bone resorption and occurs when bone resorption exceeds bone formation [2,4].

Bone modeling and remodeling are essential processes that occur during bone development, growth, and metabolism. Bone remodeling is a dynamic process by which bone that has already grown is maintained in a healthy state through bone resorption and new bone formation. Osteoclasts play a crucial role by resorbing bone and dissolving its minerals, while osteoblasts actively participate in making the new bone matrix. Thus, the process of remodeling is the result of the balanced between the actions of the osteoblasts and osteoclasts [5]. When the balance is lost, bone homeostasis is interrupted, resulting in osteoporosis or osteopetrosis.

The differentiation of myeloid progenitor cells into osteoclasts is triggered by the binding of RANKL to the receptor activator of nuclear factor -κB (RANK) on myeloid lineage cells [6]. RANKL activates nuclear factor-activated T cells c1 (NFATc1), an essential transcription factor in osteoclast differentiation via the RANK/RANKL signaling pathway. In turn, NFATc1 induces the expression of osteoclast-specific genes, including TRAP, CTSK, DC-STAMP, OSCAR, c-Fos and others involved in cell differentiation. The increased expression results in the differentiation of osteoclasts into mature osteoclasts, potentially exacerbating osteoporosis [7,8].

Barley (*Hordeum vulgare*) is a grass belonging to the family *Poaceae*, tribe *Triticeae*, and genus *Hordeum*. Barley is the fourth largest grain crop produced worldwide [9]. As the interest of consumers in nutrition and health has increased, there is rising interest in barley. Barley has been reported to have detoxification, antioxidant, and cholesterol-lowering effects as it contains functional ingredients such as polyphenol, saponarin, and flavonoids [10,11]. Barley grains are not only an excellent energy source, but also widely used in processed foods due to their unique taste and physiological benefits [12]. The Betaone barley, a naked waxy barley variety, used in this study was developed in 2015 by the Korea Rural Development Administration. Betaone was reported to have the highest beta-glucan content of about 11.4% among the Korean barley varieties developed to date [13]. In this study, the anti-osteoporosis effects of a Betaone water extract (B1W) were examined in ovariectomy-induced bone loss in mice and RANKL-induced osteoclast differentiation *in vitro*.

## Materials and methods

### Preparation of the Betaone water extract (B1W)

The Betaone barley variety used in this experiment was cultivated by the Korea Rural Development Administration, following standard cultivation methods in a test field of the National Institute of Crop Science. The barley was ground into powder using a Retsch centrifugal mill (Zm 100, Retsch GmbH &Co, Haan, Germany) equipped with a 0.2 mm sieve. The powder was defatted three times by adding a nine-fold volume of hexane and stirring at room temperature for 24 h. The mixture was filtered, and the residue was extracted with a nine-fold volume of prethanol for 24 h twice, and then filtered. Subsequently, the residue was extracted twice for 2 h with distilled water in the ratio of 1:9. The mixture was left at 4 °C for 22 h, after which only the supernatant was collected. The collected supernatant was filtered and concentrated using a vacuum rotary evaporator (N-1000, EYELA, Yokyo, Japan), and freeze-dried. 400 g of B1 powder was extracted with 3.6 L of distilled water and concentrated, and approximately 43.202 g of extracted powder was obtained. The yield of the B1W was about 10.8%. The total flavonoid and phenolic contents of B1W were analyzed by the methods described in previous papers [14].

## Measurement of bioactive substances in B1W – triphenyl hexene (TH) and ferulic acid (FA)

For the TH analysis, the crude extract was dissolved with 80% methanol and filtered through a 0.2 μm regenerated cellulose syringe filter. The sample preparation for the FA analysis was carried out using the Gamel and Abdel-Aal [15] method as follows: 15 mL of 2 N NaOH was added for alkaline extraction and stirred for 2 h. The pH was adjusted to 2 with HCl and 15 mL of diethyl ether/ethyl acetate (1:1) was added. The mixture was then shaken for 10 min and centrifuged at $140,000 \times$ g for 5 min. Then the supernatant was concentrated, redissolved in 95% ethyl alcohol, and filtered with a 0.2 μm membrane filter. The TH and FA contents were determined using an ultra-performance liquid chromatography (UPLC–UV, Waters) system at 254 nm and 280 nm, respectively. The column was a Halo C-18 column (2.7 μm, 100 mm × 2.1 mm inner diameter 90 Å), the temperature used was 35 °C, and the solvents were mobile phase A (0.1% formic acid in distilled water, v/v) and mobile phase B (100% acetonitrile) for TH and mobile phase A (0.1% TFA in distilled water, v/v) and mobile phase B (100% acetonitrile) for FA, respectively.

## Animal experimental design and ovariectomy-induced bone erosion

All procedures involving mice were conducted with strict adherence to the Sunchon National University Institutional Animal Care and Use Committee guidelines for the care and use of laboratory animals (Permit Number: SCNU IACUC-2022-06). Five-week-old female ICR mice were sourced from RaonBio (Yongin, Korea). The acclimatized mice underwent bilateral ovariectomy (n = 40) or sham surgery (n = 8) using the dorsal approach under general anesthesia with avertin (250 mg/kg). After one week of recovery, the ovariectomized (OVX) mice were randomly divided into five groups of eight mice; control group (OVX), OVX + low dose of B1W (100 mg/kg, B1W100), OVX + medium dose of B1W (200 mg/kg, B1W200), OVX + high dose of B1W (300 mg/kg, B1W300), and OVX + 17β-estradiol (0.5 mg/kg, E2). The B1W was provided by mixing it with the diet and estradiol was injected subcutaneously. Food intake and body weight were measured daily and weekly, respectively. After 6 weeks of the experiment, the mice were sacrificed by cervical dislocation, and blood was collected from the heart for biochemical analysis. The uteri were excised and weighed. The femurs were harvested and fixed in 10% neutral buffered formalin for 1 day. The fixed femurs were scanned and analyzed by using the SkyScan 1272 micro-CT imaging system provided (Bruker, Billerica, MA, USA).

## Assessment of serum biomarkers of osteoporosis

The serum C-telopeptide of collagen type I (CTX-1), osteocalcin, and estrogen levels were determined using a mouse CTX-I ELISA Kit (CSB-E12782m, CUSABIO, TX, USA), a mouse osteocalcin/bone gamma-carboxyglutamate (gla) protein ELISA kit (CSB-E06917m, CUS-ABIO), and a mouse estrogen ELISA Kit (CSB-E07280m, CUSABIO), respectively.

## Isolation of bone marrow-derived macrophage (BMM) and osteoclast differentiation

*In vitro* cell experiments were performed as described in a previously published paper [16]. BMMs were isolated from the femurs and tibiae of five-week-old male ICR mice (n = 2; Raon-Bio). The experimental protocol was approved by the Sunchon National University Institutional Animal Care and Use Committee (Permit No. SCNU IACUC 2022-06). The BMM were isolated from unfractionated bone marrow cell (BMC) cultures using a previously reported

method [16], the cells were plated at $1 \times 10^4$ cells/well in 96 well plate. The BMMs were pretreated with B1W for 1 h before culturing with RANKL (10 ng/mL; R&D Systems, Minneapolis, MN, USA) and macrophage colony-stimulating factor (M-CSF; 30 ng/mL) in 10% fetal bovine serum (FBS) α-minimum essential medium (α -MEM) for 4 days.

### Tartrate-resistant acid phosphatase (TRAP) staining assay

The differentiated cells were washed with phosphate-buffer saline and fixed with 10% formalin for 5 minutes. The fixed cells were reacted with Triton X-100 (0.1%) for 10 min and then treated with the TRAP solution (Sigma-Aldrich, St. Louis, MO, USA) for 10 min in the dark and at room temperature. The TRAP-positive multinucleated cells (nuclei = 3) were counted as mature osteoclasts.

### Cytotoxicity assay of B1W

BMM cells were seeded with $1 \times 10^4$ cells per well and cultured in 10% FBS α-MEM containing M-CSF (30 ng/mL) for 24 h at 37 °C. The next day, B1W was added to the cells and incubated for 3 days 37 °C. Cell viability was evaluated by using a Cell Counting Kit-8 (CCK-8; Tomado Molecular Technology, Kumamoto, Japan) according to the manufacturer's protocol.

### Bone pit formation assay

An experiment using an osteo assay plate (24-well plate; Corning Inc., Corning, NY, USA) was performed to measure the bone resorption activity of the BMMs. After incubation of the BMMs for 3 days in 10% FBS α-MEM containing M-CSF and RANKL to cause differentiation, they were subjected to vehicle or B1W treatment. The bone pit areas were observed under a light microscope (magnification, 50X; Leica Microsystems, Wetzlar, Germany) and measured by using the Image J software (NIH, Bethesda, MD, USA).

### Real-time qPCR

Real-time qPCR analysis was performed as previously described [16]. The BMMs were incubated with M-CSF (30 ng/mL) in 10% FBS α-MEM and activated with RANKL (10 ng/mL) for 0, 1, 2, or 3 days in the presence of B1W. The PCR primer sets (Table 1) were designed using an online primer3 program. The total RNA was isolated using TRIzol reagent (Thermo Fisher Scientific Inc., Waltham, Mass., USA), and cDNA was synthesized using the Moloney murine leukemia virus (M-MLV) cDNA synthesis kit (Enzynomics, Daejeon, Korea). The qPCR was performed using a TOPreal™ qPCR 2x PreMIX (BioRad, Hercules, CA, USA) in a real-time PCR detection system (BioRad). The mRNA levels of the genes were determined by using the $2^{-\Delta\Delta CT}$ method. Glyceraldehyde-3-phosphate dehydrogenase (*GAPDH*) was used as the internal standard.

### Western blot analysis

A western blot analysis was conducted as previously described [17]. Briefly, the BMMs were incubated with RANKL (10 ng/mL) and M-CSF (30 ng/mL) in 10% FBS α-MEM for 0, 1, 2, or 3 days in the presence of B1W. The harvested cells were lysed in a lysis buffer containing a protease inhibitor and quantified by the Bradford assay. Isolated proteins were separated using 10% sodium dodecyl-sulfate polyacrylamide gel electrophoresis (SDS-PAGE) and transferred to a polyvinylidene difluoride (PVDF membrane, Millipore, USA). The membrane was incubated with primary antibody (NFATc1, p-Jun N-terminal kinase [p-JNK], JNK, p-p38, p38) at 4 °C overnight. β-actin was used as the internal control.

**Table 1. Primer sequences used in this study.**

| Gene of interest | Sequence (5' → 3') |
|---|---|
| NFATc1 Forward | GGGTCAGTGTGACCGAAGAT |
| NFATc1 Reverse | GGAAGTCAGAAGTGGGTGGA |
| c-Fos Forward | CCAGTCAAGAGCATCAGCAA |
| c-Fos Reverse | AAGTAGTGCAGCCCGGAGTA |
| CTSK Forward | GGCCAACTCAAGAAGAAAAC |
| CTSK Reverse | GTGCTTGCTTCCCTTCTGG |
| DC-STAMP Forward | CCAAGGAGTCGTCCATGATT |
| DC-STAMP Reverse | GGCTGCTTTGATCGTTTCTC |
| OSCAR Forward | CTGCTGGTAACGGATCAGCTC |
| OSCAR Reverse | CCAAGGAGCCAGAACCTT |
| TRAP Forward | GATGACTTTGCCAGTCAGCA |
| TRAP Reverse | ACATAGCCCACACCGTTCTC |
| GAPDH Forward | AACTTTGGCATTGTGGAAGG |
| GAPDH Reverse | ACACATTGGGGGTAGGAACA |

## Statistical analysis

All data are presented as the means ± standard error. Statistical analysis was performed using the SPSS version 26 software (SPSS Inc., Chicago, IL, USA). Differences between the groups were analyzed using the student's *t*-test. Probability (P) values < 0.05 were considered statistically significant.

## Results

### Phytochemical contents of B1W

The total phenolic and flavonoid contents in B1W were $4.80 \pm 0.001$ mg/g and $0.812 \pm 0.004$ mg/g, respectively. TH and FA were detected as the most abundant substances at $1.60 \pm 0.008$ mg/g and $0.206 \pm 0.003$ mg/g, respectively. The β-glucan content of the B1W was $5.10 \pm 0.19\%$ (Table 2). TH and FA are phytochemicals known to inhibit osteoclast formation, as reported in previous studies [17,18]. In this study, we identified TH and FA in B1W using UPLC-UV. The standards of TH and FA were observed at retention times of 47.6 minute and 12.5 minute, respectively. TH and FA were detected at same retention times in B1W (Fig 1).

### B1W inhibited ovariectomy-induced osteoporosis in mice

Induction of bone loss through ovariectomy in mice is a well-established and widely used model [19,20,21]. In the current study, we used OVX mouse model to evaluate the inhibiting effect of B1W on osteoporosis *in vivo*. The trabecular bone of the femur was reduced in

**Table 2. Phytochemical contents of B1W.**

| Triphenyl hexene (mg/g) | Ferulic acid (mg/g) | β-Glucan (%) | Total polyphenol (mg/g)[1] | Total flavonoid (mg/g)[2] |
|---|---|---|---|---|
| $1.60 \pm 0.008$ | $0.206 \pm 0.003$ | $5.10 \pm 0.19$ | $4.80 \pm 0.001$ | $0.812 \pm 0.004$ |

All experiments were performed with three replications. Data are presented as mean ± SD.

[1]Total phenolic acid content was quantified according to a calibration standard curve of gallic acid.

[2]Total flavonoid content was quantified according to a calibration standard curve of catechin hydrate.

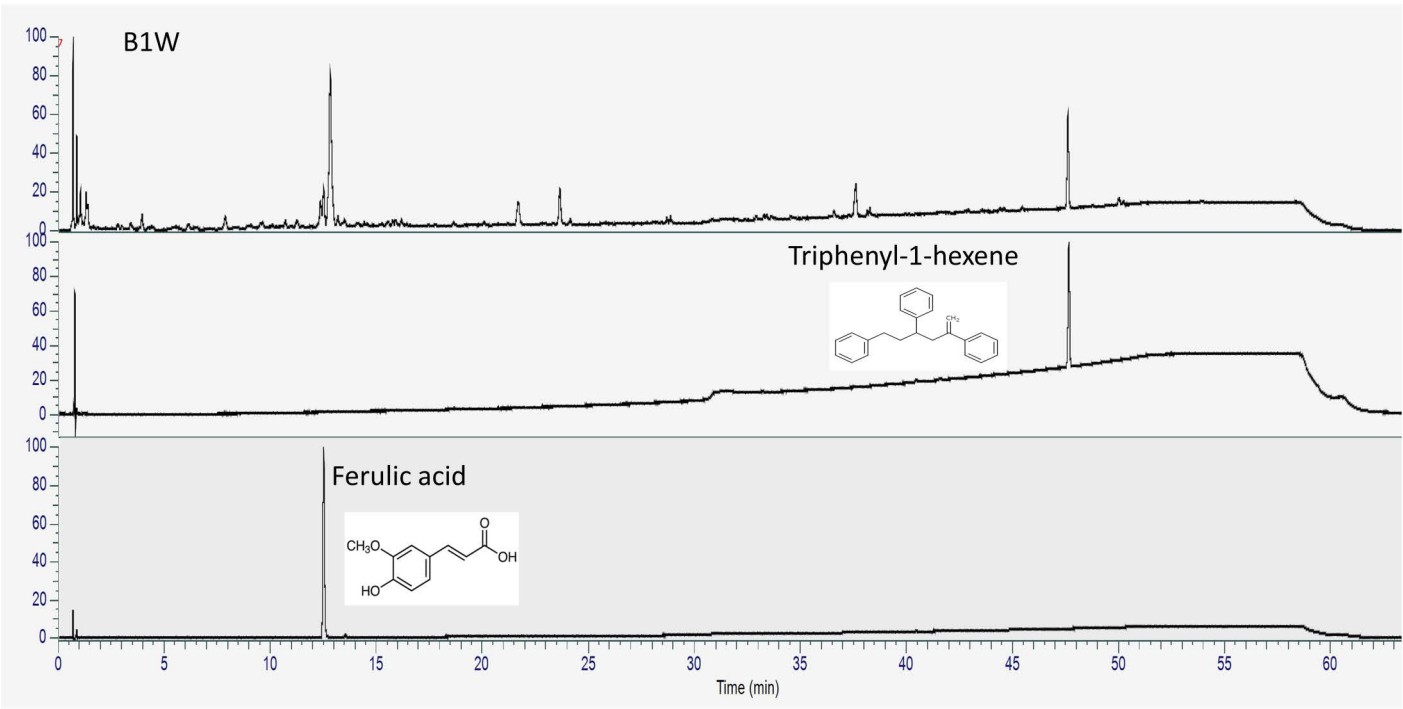

**Fig 1. HPLC chromatogram of B1W, Triphenyl-1-hexane and Ferulic acid standard solution.**

the OVX mice compared to that in the sham-operated mice. However, B1W-supplemented groups and estrogen-administered group showed a reduction in OVX-induced osteoporosis and an increase bone mineral density (BMD) and bone volume fraction (BV/TV) in a dose-dependent manner compared to the OVX group (Fig 2a,b). In addition, significance, B1W (300 mg/kg) and E2 significantly decreased trabecular separation (Tb. Sp), which was increased due to the aggravation of osteoporosis compared to the OVX group (Fig 2b).

## B1W suppressed ovariectomy-induced bone resorption and weight gain

The serum CTX-1 levels, an indicator of bone resorption, were increased in the OVX group compared to the sham group. However, B1W supplementation decreased the CTX-1 levels compared to the OVX group, with a particularly notable reduction observed in the B1W300 group, where levels were decreased by 34%. However, these results were not statistically significant (Fig 3a). The serum osteocalcin level did not vary significantly among the groups (Fig 3a). The serum estrogen level was significantly increased in the B1W200, B1W300, and E2 groups compared to the OVX group (Fig 3a).

In the OVX group, the uterus underwent degeneration due to the removal of the ovaries, resulting in rapid weight loss. However, the B1W200, B1W300 and E2 groups suppressed uterine degeneration compared to the OVX group (Fig 3b). B1W did not affect food intake and body weight gain in OVX mice (Fig 3b).

## B1W suppressed RANKL-induced osteoclast differentiation

Osteoporosis results from the differentiation of osteoclasts into multinucleated cells via a RANKL-mediated reaction. To investigate the inhibitory effect of B1W on osteoclast differentiation, B1W (0, 1, 3, 10, or 30 μg/mL) was added to the BMMs treated with M-CSF prior to

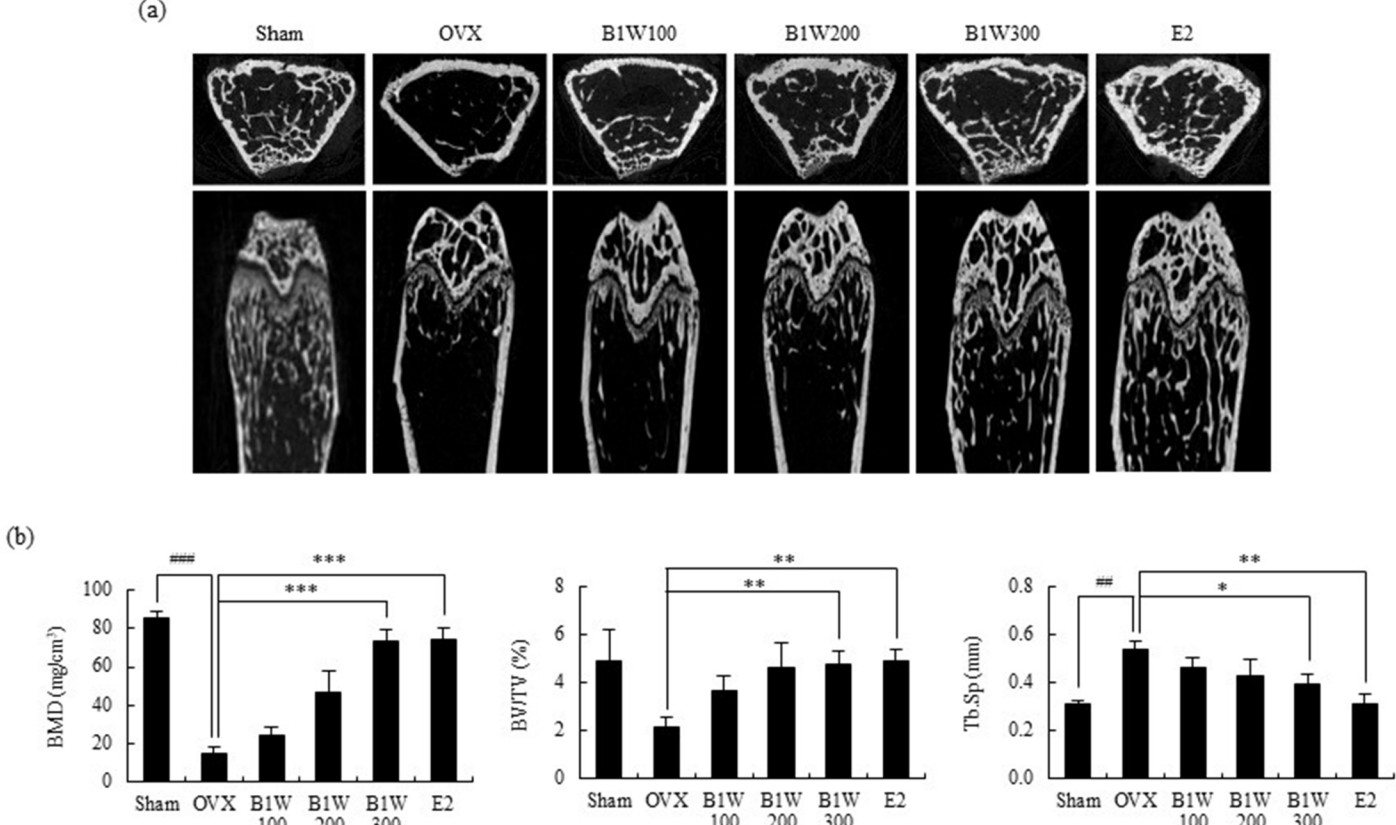

**Fig 2. The effects of B1W on (a) Micro-CT, (b) bone mineral density (BMD), bone volume fraction (BV/TV), and trabecular separation (Tb.Sp) of femurs.** Data are presented as the mean ± SEM. $^{\#\#}P < 0.01$, $^{\#\#\#}P < 0.001$ versus the Sham group; $^{*}P < 0.05$, $^{**}P < 0.01$, $^{***}P < 0.001$, versus the OVX group.

the RANKL treatment to induce osteoclast differentiation. After 4 days of culture, the RANKL-treated BMMs differentiated into TRAP (+)-multinucleated cells, but the level of differentiation was significantly reduced by the addition of B1W (Fig 4a). The number of TRAP (+) cells with 3 or more nuclei, characteristic of differentiated osteoclasts, was significantly reduced at B1W concentrations from 1 to 30 μg/mL (Fig 4b). To determine whether the suppression of osteoclast differentiation was due to the cytotoxicity of B1W, a cytotoxicity study was conducted to assess whether the growth of the BMM cultures was affected by B1W. B1W did not exhibit a cytotoxic effect on BMMs at the concentrations (1-30 μg/mL) used in this study. The 1, 3, 10, and 30 μg/mL B1W concentration increased BMM survival, but the increases were not significant (Fig 4c).

## B1W reduced the RANKL-mediated bone resorptive activity of osteoclasts

This study investigated the effect of B1W on the bone resorption function of mature osteoclasts. Mature osteoclasts typically create a large pit area due to their bone resorption ability, but the B1W treatment significantly reduced the pit area in a dose-dependent manner (Fig 4d).

## B1W downregulated RANKL-mediated osteoporosis-related gene expression

To investigate the mechanistic role of B1W in the inhibition of osteoclast differentiation, we examined its effects on the activation of osteoporosis-related genes. The mRNA expression

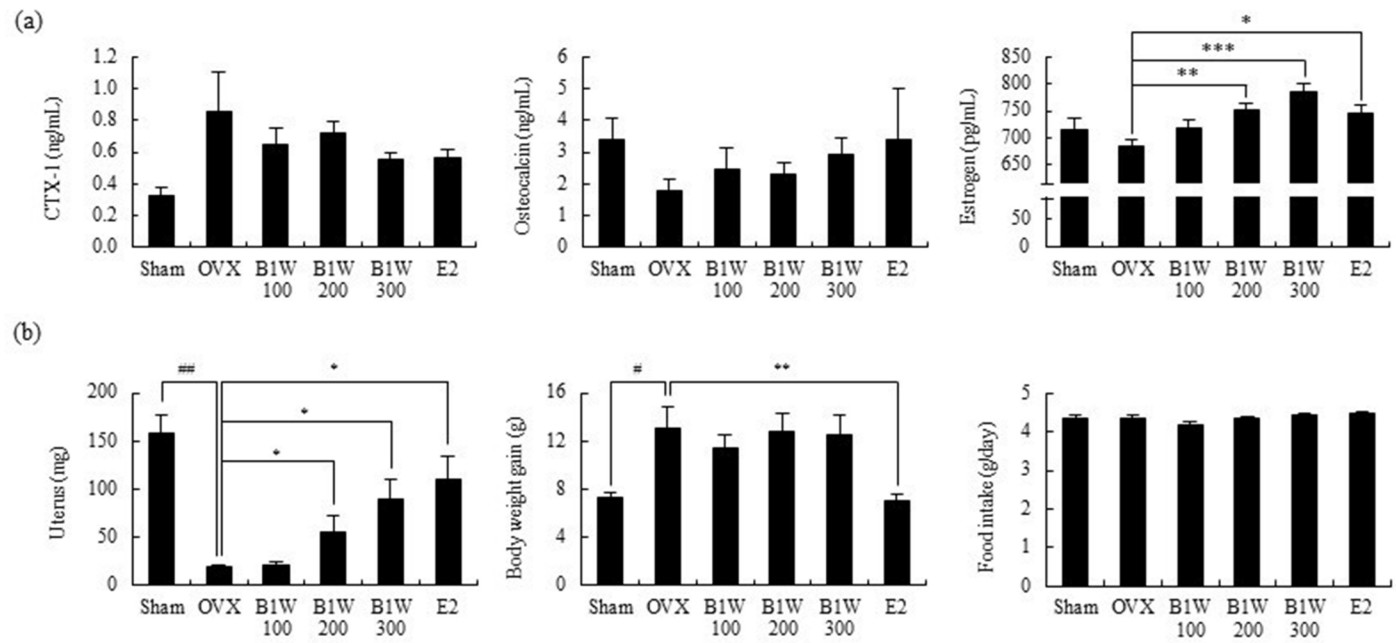

**Fig 3. The effect of B1W on (a) serum biomarkers, and (b) weight of uterus, body weight gain and food intake in OVX mice.** Data are presented as the mean ± SEM. #$P < 0.05$, ##$P < 0.01$, versus the Sham group; *$P < 0.05$, **$P < 0.01$, ***$P < 0.001$, versus the OVX group.

of *NFATc1* in the RANKL-only treated cells (control) increased over the three-day treatment period. However, that expression was significantly decreased in the B1W-treated cells compared to the control cells. Similarly, B1W significantly downregulated the mRNA expression of the osteoporosis-related genes c-Fos, TRAP, CTSK, DC-STAMP and OSCAR (Fig 5).

## B1W inhibited the RANKL-induced protein expression of NFATc1

A western blot analysis was performed to examine the effect of B1W on the expression of NFATc1, a major protein related to RANKL-induced osteoclast differentiation. The translation expression level of NFATc1 was increased without B1W. The expression of NFATc1 was the highest at 2 days. However, the expression of NFATc1 was significantly decreased by treatment with 10 μg/mL of B1W (Fig 6), indicating that B1W inhibits NFATc1 expression and osteoclast formation.

## B1W inhibited RANKL-induced phosphorylation of MAP kinases

MAPK signaling, an important process in osteoclast differentiation, was investigated through the phosphorylation of p38 and JNK in BMMs after inducing osteoclast differentiation with RANKL using the western blot method. Analysis of the results showed that B1W significantly reduced the RANKL-induced phosphorylation levels of p38 and JNK (Fig 7).

## Discussion

The bioactive components of plant extracts, such as polyphenols and flavonoids, have been shown to function as inhibitors of osteoporosis and can be helpful in the treatment of osteoporosis [22]. In this study, the total phenolic and flavonoid contents of B1W were 4.80 ± 0.001 mg/g and 0.812 ± 0.004 mg/g, respectively, which were higher than the content found in the extracts of other grains such as oats [23]. In an earlier study, we identified

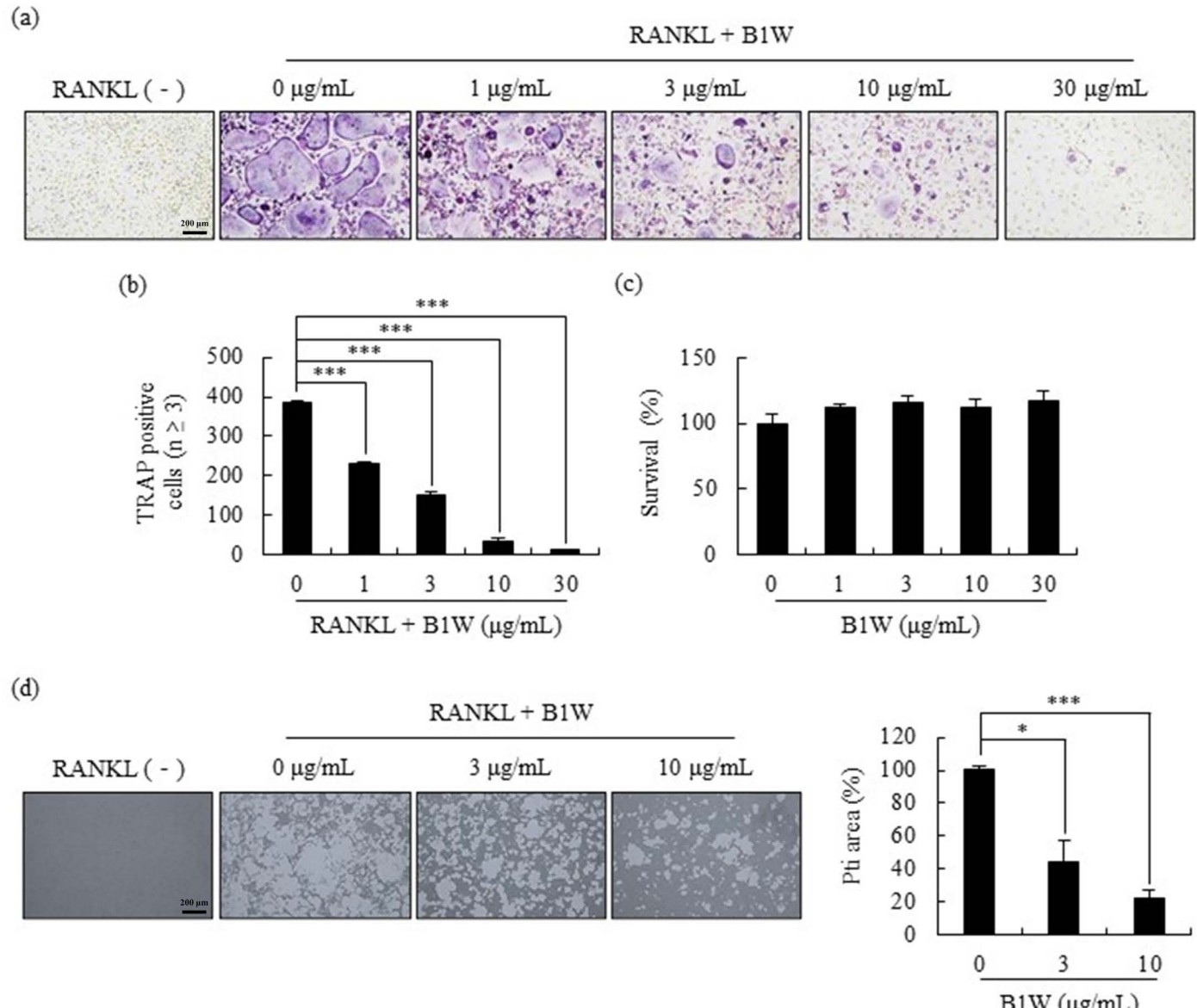

**Fig 4. The effect of B1W on (a) osteoclast differentiation, (b) TRAP-positive multinucleated cells (3 or more nuclei), (c) viability of BMM, and (d) pits area.** Data from three independent experiments are presented as the mean ± SEM values. Pit areas were quantified using the Image J program. *P < 0.05, ***P < 0.001, versus the 0 µg/mL B1W group.

TH and FA as the two major active compounds of B1W. The TH and FA contents were 159.59 ± 0.836 mg/100 g, 20.57 ± 0.273 mg/100 g, respectively. The study also reported that TH inhibits osteoporosis *in vitro* and *in vivo* [17]. Therefore, this study further investigated the anti-osteoporosis effects of B1W in RANKL-induced osteoclastogenesis and OVX-induced bone loss.

The present study demonstrated that B1W (300 mg/kg/day) effectively improved bone morphology, BMD, BV/TV, and Tb. Sp compared to the OVX control. Osteoporosis is a skeletal disease that increases the risk of fracture due to a reduction in bone strength as a result of a loss in bone mass and bone quality [24]. Bone mass is expressed as BMD, while bone

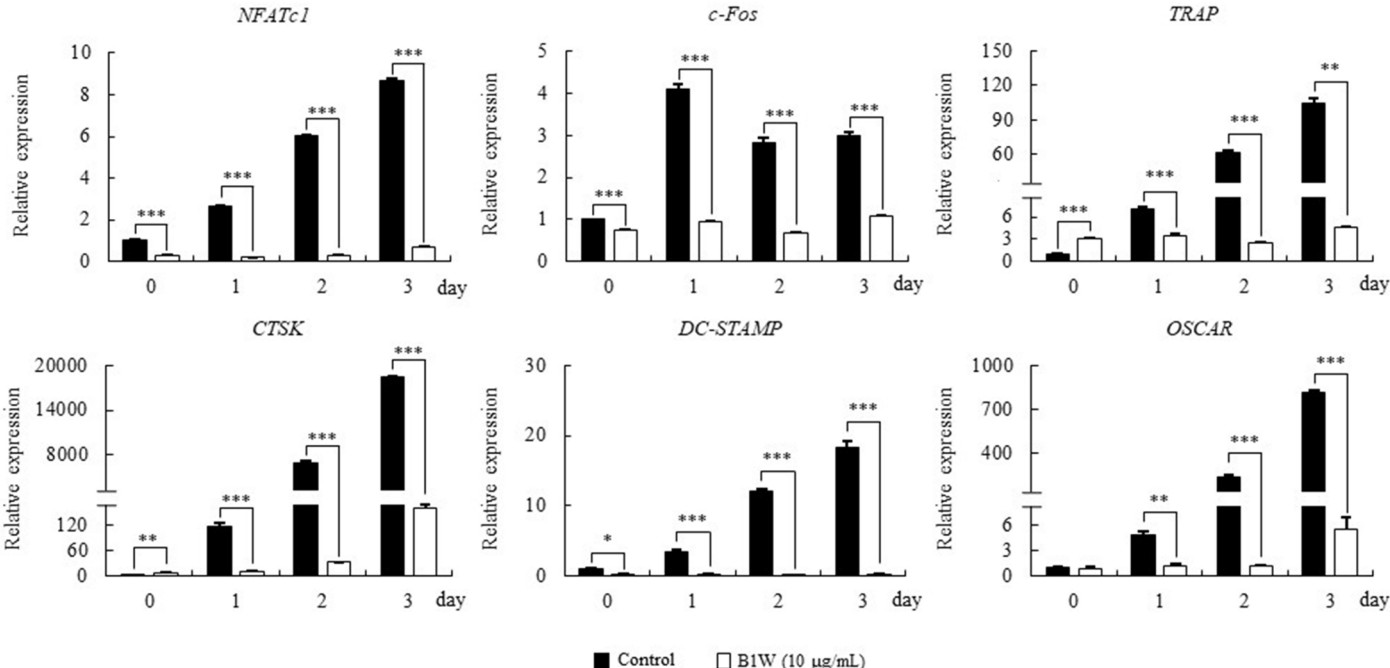

**Fig 5. The effects of B1W on RANKL-mediated mRNA expression of osteoporosis-related genes.** Glyceraldehyde-3-phosphate dehydrogenase (GAPDH) was used as the internal control. Data from three independent experiments are presented as the mean ± SEM values. *$P < 0.05$, **$P < 0.01$, ***$P < 0.001$, versus the control group.

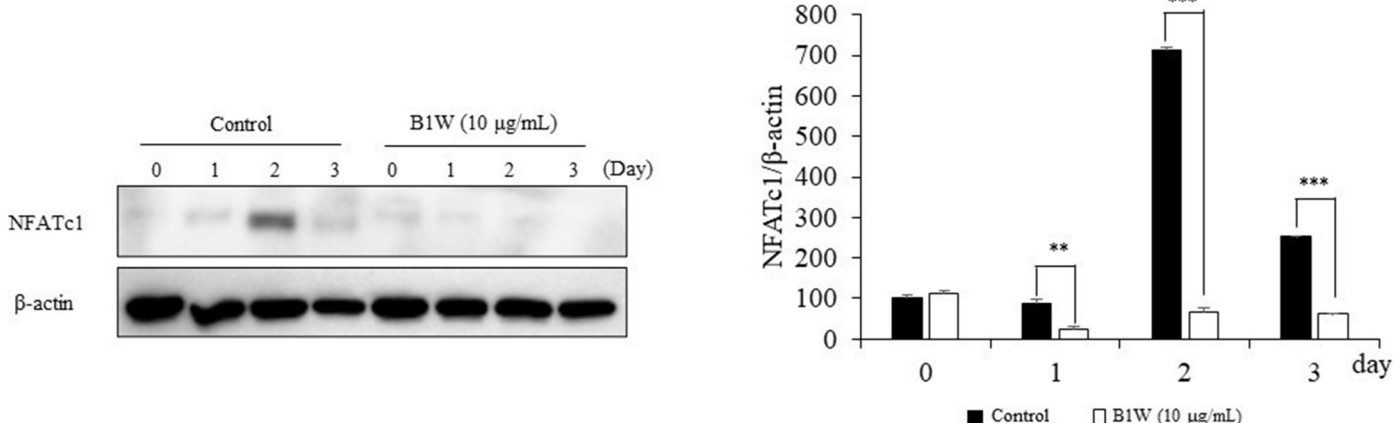

**Fig 6. The effects of B1W on the expression of NFATc1 protein under RANKL.** The band was calculated based on the β-actin loading amount. Data from three independent experiments are presented as the mean ± SEM values. **$P < 0.01$, ***$P < 0.001$, versus the control group.

quality is reflected by bone structure, bone turnover rate, mineralization, and accumulation of micro-damage. To reduce the risk of bone fractures, it is crucial to not only increase bone density but also to improve bone quality. BV/TV reflects trabecular bone connectivity. Improved BV/TV by B1W supplementation is beneficial as improved trabecular bone connectivity is physiologically beneficial in terms of bone quality [25]. Furthermore, the serum CTX-1 level, a marker of bone resorption, in the B1W300 group, showed a tendency to decrease compared to that of the OVX group. During bone resorption, short portions of the C-terminus of collagen

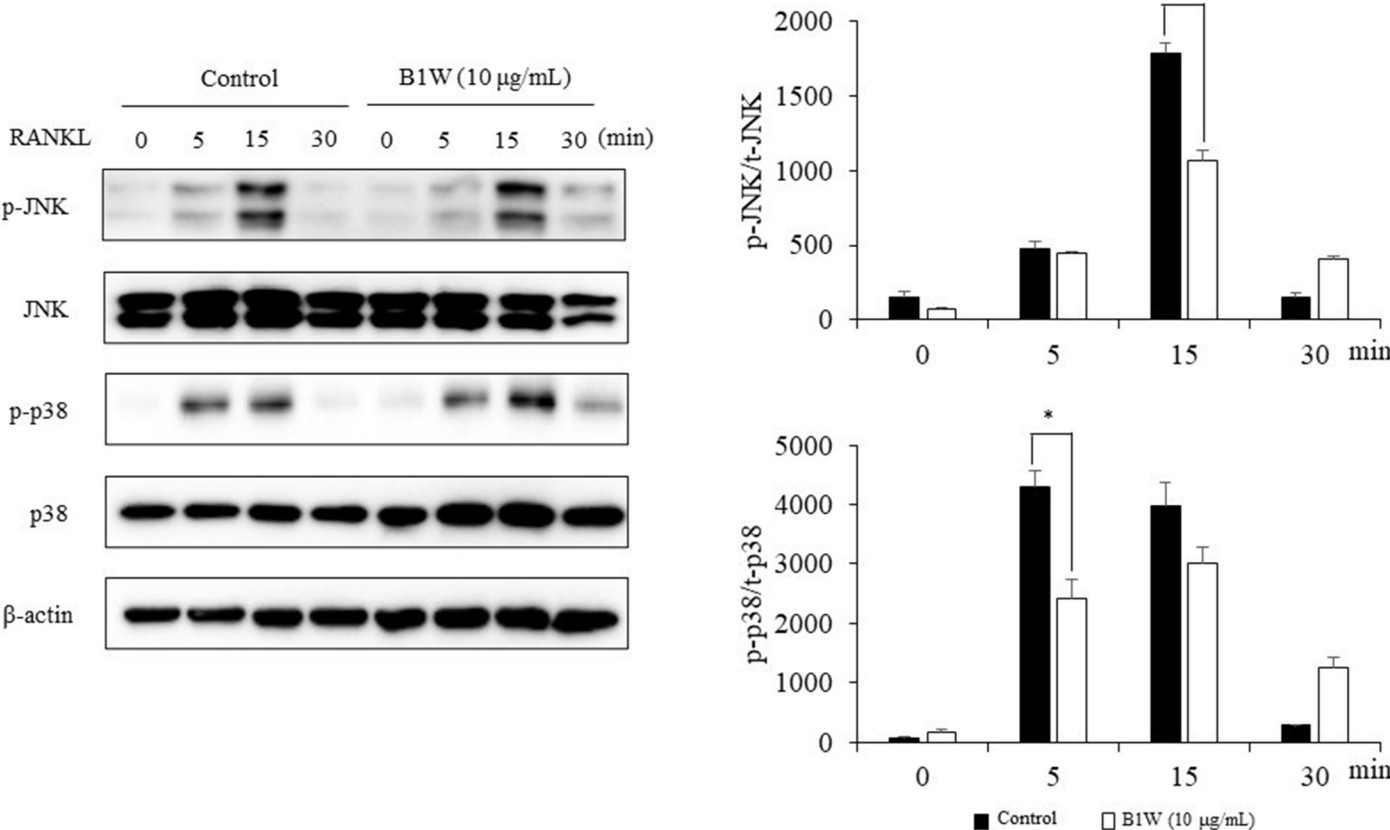

**Fig 7. The effects of B1W on MAP kinase signaling were analyzed by western blotting.** The phosphorylated and total forms of p38 and JNK were measured. All data were normalized by β-actin. Data from three independent experiments are presented as the mean ± SEM. *$P < 0.05$, **$P < 0.01$, versus the control group.

molecules are degraded by osteoclasts and released into the blood. Therefore, CTX-1 serve as a marker of osteoclastic activity [26]. Thus, the study results suggest that B1W can ameliorate OVX-induced bone loss and quality and suppress bone resorption.

An increase in osteoclastogenesis can lead to osteoporosis. The processes related to osteoclast differentiation, activation, and death have been previously identified, and RANKL, a cytokine produced and secreted by osteoblasts (or activated immune cells), plays a decisive role in promoting the differentiation and activity of osteoclasts by binding to receptors located in the osteoclasts [27]. The osteoclast differentiation is a multi-step process, mediated by cell signaling triggered by RANKL-RANK binding. Specifically, RANKL-RANK signal transduction leads to osteoclast formation through related signaling processes and is a major contributor to pathological bone loss in osteoporosis. Furthermore, elucidation of such signaling is crucial for the development of targeted treatments. The RANKL-RANK binding results in the activation of c-Fos-NFATc1, an essential transcription factor in osteoclast differentiation, and rapid activation of MAPK signaling molecules such as p38 and JNK [28]. Therefore, this study investigated the effect of B1W on RANKL-induced osteoclastogenesis to understand its underlying mechanism of action.

The treatment with B1W inhibited osteoclast formation by significantly reducing the differentiation of RANKL-treated macrophages into osteoclasts. Additionally, B1W demonstrated no cytotoxic effects on the macrophages. The NFAT family plays a critical role as transcription

factors involved in the regulation of various biological systems, including osteoclast formation [29]. Therefore, the expression levels of NFATc1 mRNA and protein were examined using real-time qPCR and western blotting, respectively, to further investigate the effect of B1W on osteoclast differentiation. B1W significantly downregulated the expression levels of RANKL-induced NFATc1 mRNA and protein during osteoclast differentiation. In addition, the mRNA expressions of c-FOS, TRAP, DC-STAMP, OSCAR and CTSK, which are markers of osteoclast differentiation and activation, were significantly downregulated by the B1W treatment. This study investigated the cellular MAP kinase pathways, including p38 and JNK, involved in osteoclast differentiation [30]. In this study, B1W inhibited RANKL-mediated phosphorylation of p38 and JNK. Consequently, B1W downregulated RANKL-mediated MAPK activation, leading to the inhibition of NFATc1 expression. The pit resorption assay, a method commonly utilized to investigate osteoclast-mediated bone resorption [31], revealed that B1W effectively inhibited RANKL-induced osteoclast resorption. These results support the results on the effectiveness of B1W in inhibiting the release of CTX, an indicator of bone resorption, in OVX mice. Previous studies have reported that TH, FA, and β-glucan, major compounds of B1W, exhibit anti-osteoporotic effects *in vitro* or *in vivo* [16,18,32]. Based on these results, the inhibitory activity of B1W on the differentiation and resorption of osteoclasts appears to be related to the presence of these phytochemical compounds in B1W.

In conclusion, the findings from this study suggest that B1W has the potential to prevent osteoclast differentiation by downregulating the expression of NFATc1 through the MAPK pathway. The suppression of key factors associated with osteoclast formation ultimately leads to the prevention of OVX-induced osteoporosis. Hence, B1W could find application as a functional food or a therapeutic agent for the treatment of bone diseases, such as osteoporosis.

## Supporting information

**S1 Fig. Raw images.**
(PDF)

## Author contributions

**Conceptualization:** Yongjin Lee, Mi-Ja Lee, Mi-Kyung Lee, Young-Jin Son.

**Data curation:** Yongjin Lee, Hyun-Jin Lee, Kwang-Jin Kim, Han-Byeol Shin, Yoon-A Shin, Holim Jin, Ju Ri Ham, Soo-Young Choi, Mi-Ja Lee, Mi-Kyung Lee, Young-Jin Son.

**Funding acquisition:** Mi-Ja Lee, Mi-Kyung Lee, Young-Jin Son.

**Investigation:** Mi-Ja Lee, Mi-Kyung Lee, Young-Jin Son.

**Project administration:** Mi-Ja Lee, Mi-Kyung Lee, Young-Jin Son.

**Resources:** Mi-Ja Lee, Mi-Kyung Lee, Young-Jin Son.

**Validation:** Yongjin Lee, Mi-Ja Lee, Mi-Kyung Lee, Young-Jin Son.

**Writing – original draft:** Yongjin Lee, Hyun-Jin Lee, Mi-Ja Lee, Mi-Kyung Lee, Young-Jin Son.

**Writing – review & editing:** Yongjin Lee, Hyun-Jin Lee, Kwang-Jin Kim, Han-Byeol Shin, Yoon-A Shin, Holim Jin, Ju Ri Ham, Soo-Young Choi, Mi-Ja Lee, Mi-Kyung Lee, Young-Jin Son.

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
