## [Decision Letter · Decision Letter 0]

30 Oct 2024

PONE-D-24-41839“Betaone” barley water extract suppresses ovariectomy-induced osteoporosis in vivo and RANKL-induced osteoclast differentiation in vitroPLOS ONE

Dear Dr. Son,

Thank you for submitting your manuscript to PLOS ONE. After careful consideration, we feel that it has merit but does not fully meet PLOS ONE’s publication criteria as it currently stands. Therefore, we invite you to submit a revised version of the manuscript that addresses the points raised during the review process.

We look forward to receiving your revised manuscript.

Kind regards,

Liangliang Xu

Academic Editor

PLOS ONE

Journal Requirements:

“This work was supported by the Cooperative Research Program for Agriculture Science and Technology Development (Grant No. PJ013524022022).”

“The authors declare that they have no conflict of interest.”

5. We noted in your submission details that a portion of your manuscript may have been presented or published elsewhere. [My research manuscript was uploaded as a preprint in research square. (https://doi.org/10.21203/rs.3.rs-2348694/v1) ] Please clarify whether this [conference proceeding or publication] was peer-reviewed and formally published. If this work was previously peer-reviewed and published, in the cover letter please provide the reason that this work does not constitute dual publication and should be included in the current manuscript.

6. We note that your Data Availability Statement is currently as follows: [All relevant data are within the manuscript and its Supporting Information files.]

Please confirm at this time whether or not your submission contains all raw data required to replicate the results of your study. Authors must share the “minimal data set” for their submission. PLOS defines the minimal data set to consist of the data required to replicate all study findings reported in the article, as well as related metadata and methods (https://journals.plos.org/plosone/s/data-availability#loc-minimal-data-set-definition ).

If your submission does not contain these data, please either upload them as Supporting Information files or deposit them to a stable, public repository and provide us with the relevant URLs, DOIs, or accession numbers. For a list of recommended repositories, please see https://journals.plos.org/plosone/s/recommended-repositories .

7. PLOS ONE now requires that authors provide the original uncropped and unadjusted images underlying all blot or gel results reported in a submission’s figures or Supporting Information files. This policy and the journal’s other requirements for blot/gel reporting and figure preparation are described in detail at https://journals.plos.org/plosone/s/figures#loc-blot-and-gel-reporting-requirements and https://journals.plos.org/plosone/s/figures#loc-preparing-figures-from-image-files . When you submit your revised manuscript, please ensure that your figures adhere fully to these guidelines and provide the original underlying images for all blot or gel data reported in your submission. See the following link for instructions on providing the original image data: https://journals.plos.org/plosone/s/figures#loc-original-images-for-blots-and-gels.   

Reviewers' comments:

Reviewer's Responses to Questions

**Comments to the Author**

1. Is the manuscript technically sound, and do the data support the conclusions?

Reviewer #1: Partly

Reviewer #2: Yes

Reviewer #3: Partly

2. Has the statistical analysis been performed appropriately and rigorously? 

Reviewer #1: I Don't Know

Reviewer #2: Yes

Reviewer #3: Yes

3. Have the authors made all data underlying the findings in their manuscript fully available?

Reviewer #1: No

Reviewer #2: Yes

Reviewer #3: No

4. Is the manuscript presented in an intelligible fashion and written in standard English?

Reviewer #1: Yes

Reviewer #2: Yes

Reviewer #3: Yes

5. Review Comments to the Author

Reviewer #1: 93-96: The yield of B1W was approximately 10.8%. Please provide the extraction yield report for B1W.

99-112: Please include ultra-performance liquid chromatography data for TH and FA, not just graphs.

117-120: Were the rats 5 weeks old at the time of modeling? Female rats at this age are not yet sexually mature, which may not be the optimal time for ovariectomy.

125-135: The experimental animals were sacrificed 6 weeks post-surgery, and serum was extracted for bioanalysis. What method was used to collect the blood? Is it venous blood?

139-147: BMMSCs were extracted from 5-week-old male rats, which may be somewhat older, potentially affecting the activity of rat bone marrow cells. How were BMMs pretreated with B1W? What passage number were the subsequent BMMSCs used for cell experiments?

139-147: A 96-well plate was seeded with 10^4 cells and cultured for four days. Is there sufficient space for growth?

157-160: In the cytotoxicity experiments, was 10 ng/ml RANKL used to induce osteoclasts for four days?

182：In Table 1, what does "sense" and "anti-sense" refer to? The formatting appears incorrect.

207: Table 2 is not formatted correctly.

273-277: The timing and dosage of RANKL induction were not annotated, nor are they indicated in the corresponding Fig. 5. What is the difference between bone pit formation and TRAP staining? The results of tissue section staining for osteoporosis should be supplemented for verification.

WB Fig. 5: The gray values are not labeled, and there appears to be a significant difference in the Day 2 group. Is there any duplication in the data?

Fig. 3: TRAP-stained positive cells should appear red, but the coloration in the article appears biased. Additionally, no scale bars are marked in panels A and D.

Reviewer #2: 123：The referenced study primarily investigates the effects of 17α estradiol, prompting consideration for replacing literature that examines 17β estradiol. This substitution would enhance the experimental rationale surrounding the use of 17β estradiol.

232、233：It is possible that the designation of "Fig. 2a" instead of "Fig. 1a" in reference to the picture could be a clerical error; please take note of this correction.

274：Based on the experimental results from WB, the statement that the expression level of NFATc1 in the control group increases over time is not entirely accurate.

Reviewer #3: The study by Yongjin Lee et al. explored effects of B1W on ovariectomy (OVX)-induced bone loss in vivo and osteoclastogenesis in vitro. The paper is well-written and presents promising results. However, there are a couple of concerns to make this study more convincing:

Please add associated biomechanical testing data to double confirm B1W-induced anti-osteoporosis.

Please supplement relevant data on effects of B1W on osteogenesis and bone formation.

6. PLOS authors have the option to publish the peer review history of their article (what does this mean? ). If published, this will include your full peer review and any attached files.

**Do you want your identity to be public for this peer review?** For information about this choice, including consent withdrawal, please see our Privacy Policy .

Reviewer #1: No

Reviewer #2: No

Reviewer #3: No

---

## [Author Response · Author response to Decision Letter 1]

17 Dec 2024

Dear Editor:

Please find enclosed our manuscript entitled ““Betaone” barley water extract suppresses ovariectomy-induced osteoporosis in vivo and RANKL-induced osteoclast differentiation in vitro” (PONE-D-24-41839) which we request you to consider for publication as an Original Article in PLOS ONE.

Reviewer #1 : Comments and Suggestions for Authors

93-96: The yield of B1W was approximately 10.8%. Please provide the extraction yield report for B1W.

Ans.) We added the extraction yield in the Materials and Methods. (400g of B1 powder was extracted with 3.6 L of distilled water and concentrated, and approximately 43.202 g of extracted powder was obtained. The yield of the B1W was about 10.8%., Line 95-97)

99-112: Please include ultra-performance liquid chromatography data for TH and FA, not just graphs.

Ans.) I added UPLC chromatogram(Figure 1) and results(Line 207-211) as you advised.

117-120: Were the rats 5 weeks old at the time of modeling? Female rats at this age are not yet sexually mature, which may not be the optimal time for ovariectomy.

Ans.) We purchased 5-week-old female ICRs and acclimated them for 1 week before performing the OVX surgery. The 6-week-old mice have well-developed ovaries, and we considered that there is no problems with the experiments. According to the mouse and human age comparisons (Dutta S, Sengupta P. Men and mice: Relating their ages. Life Sci . 2016;152:244-248. doi:10.1016/j.lfs.2015.10.025):

A 1-week-old mouse is roughly equivalent to a 1-year-old human.

A 6-week-old mouse corresponds to an 18- to 20-year-old human and is considered sexually mature.

A 1-year-old mouse is approximately equivalent to a 30- to 40-year-old human.

A 2-year-old mouse is comparable to a 70- to 80-year-old human and is considered elderly.

Also, other researchers performed the experiment with 6-week-old OVX model mice below references.

[1) Govoni KE, Wergedal JE, Chadwick RB, Srivastava AK, Mohan S. Prepubertal OVX increases IGF-I expression and bone accretion in C57BL/6J mice. Am J Physiol Endocrinol Metab. 2008;295(5):E1172-E1180. doi:10.1152/ajpendo.90507.2008.

2) Kitajima Y, Ogawa S, Egusa S, Ono Y. Soymilk Improves Muscle Weakness in Young Ovariectomized Female Mice. Nutrients. 2017;9(8):834. Published 2017 Aug 4. doi:10.3390/nu9080834]

125-135: The experimental animals were sacrificed 6 weeks post-surgery, and serum was extracted for bioanalysis. What method was used to collect the blood? Is it venous blood? - heart extraction

Ans.) We obtained the sufficient blood samples for physiological analysis using a heart extraction, and we added this procedure to the Materials and Methods section(Line 127-129).

139-147: BMMSCs were extracted from 5-week-old male rats, which may be somewhat older, potentially affecting the activity of rat bone marrow cells. How were BMMs pretreated with B1W? What passage number were the subsequent BMMSCs used for cell experiments?

Ans.) B1W was treated at the next day after BMMSCs seeding. We performed in vitro cell study using the enough quantities of cells (40×106). The passage number of BMMSCs used was 1.

157-160: In the cytotoxicity experiments, was 10 ng/ml RANKL used to induce osteoclasts for four days?

Ans.) We performed the cytotoxicity experiments without RANKL treatment. We identified the cytotoxic effect during the proliferation of BMMs treated M-CSF. In the osteoclastogenesis research, the cytotoxicity experiments are usually used using BMMs without RANKL treatment like below references.

[1]Wu X, Huang L, Liu J. Effects of adiponectin on osteoclastogenesis from mouse bone marrow-derived monocytes. Exp Ther Med. 2019;17(2):1228-1233. doi:10.3892/etm.2018.7069,

2) Qu Y, Liu X, Zong S, Sun H, Liu S, Zhao Y. Protocatechualdehyde Inhibits the Osteoclast Differentiation of RAW264.7 and BMM Cells by Regulating NF- κ B and MAPK Activity [retracted in: Biomed Res Int. 2023 Nov 29;2023:9820815. doi: 10.1155/2023/9820815]. Biomed Res Int . 2021;2021:6108999. Published 2021 Jul 16. doi:10.1155/2021/6108999,

3) Chen S, Chu B, Chen Y, et al. Neferine suppresses osteoclast differentiation through suppressing NF-κB signal pathway but not MAPKs and promote osteogenesis. J Cell Physiol . 2019;234(12):22960-22971. doi:10.1002/jcp.28857,

4) Zhang S, Huo S, Li H, et al. Flufenamic acid inhibits osteoclast formation and bone resorption and act against estrogen-dependent bone loss in mice. Int Immunopharmacol . 2020;78:106014. doi:10.1016/j.intimp.2019.106014]

182：In Table 1, what does "sense" and "anti-sense" refer to? The formatting appears incorrect.

Ans.) As you advised, I corrected the terms, forward and reverse instead of sense and anti-sense.

207: Table 2 is not formatted correctly.

Ans.) We modified as advised.

273-277: The timing and dosage of RANKL induction were not annotated, nor are they indicated in the corresponding Fig. 5.

Ans.) We described the timing and dosage of RANKL to differentiate osteoclast in Materials and Methods (Line 187-189).

What is the difference between bone pit formation and TRAP staining? The results of tissue section staining for osteoporosis should be supplemented for verification.

Ans.) Bone pit assays evaluate the functional activity of osteoclasts by measuring their bone resorption capacity. Meanwhile, we can identify the osteoclasts and visualize TRAP-positive multinucleated cells. So, we think the confirmation of the resorption activity of bone pit assay and the identification of TRAP stained osteoclasts is enough to provide the activity of B1W against osteoclastogenesis. We are going to plan to perform the tissue section staining for osteoporosis for the double-check verification.

Fig. 3: TRAP-stained positive cells should appear red, but the coloration in the article appears biased. Additionally, no scale bars are marked in panels A and D.

Ans.) The color of TRAP-stained positive cells is seen in red or similar to violet. The color of our result was similar to violet. According to other reference, the color is similar to ours like below.

And the scale bars was inserted in the figure 4 panels (a) and (d).

(Gu JH, Tong XS, Chen GH, et al. Regulation of matrix metalloproteinase-9 protein expression by 1α, 25-(OH)₂D₃ during osteoclast differentiation. J Vet Sci. 2014;15(1):133-140. doi:10.4142/jvs.2014.15.1.133)

(Chevalier C, Çolakoğlu M, Brun J, et al. Primary mouse osteoblast and osteoclast culturing and analysis. STAR Protoc. 2021;2(2):100452. Published 2021 Apr 13. doi:10.1016/j.xpro.2021.100452)

Reviewer #2: Comments and Suggestions for Authors

123：The referenced study primarily investigates the effects of 17α estradiol, prompting consideration for replacing literature that examines 17β estradiol. This substitution would enhance the experimental rationale surrounding the use of 17β estradiol.

Ans.) I added references (reference number 20, 21) as you advised.

232、233：It is possible that the designation of "Fig. 2a" instead of "Fig. 1a" in reference to the picture could be a clerical error; please take note of this correction.

Ans.) I corrected it as comment (Line 238-239).

274：Based on the experimental results from WB, the statement that the expression level of NFATc1 in the control group increases over time is not entirely accurate.

Ans.) I corrected the mistyping in the sentence (Line 280-282).

Reviewer #3: Comments and Suggestions for Authors

The study by Yongjin Lee et al. explored effects of B1W on ovariectomy (OVX)-induced bone loss in vivo and osteoclastogenesis in vitro. The paper is well-written and presents promising results. However, there are a couple of concerns to make this study more convincing: Please add associated biomechanical testing data to double confirm B1W-induced anti-osteoporosis. Please supplement relevant data on effects of B1W on osteogenesis and bone formation.

Ans.) We performed the experiment for osteoblast activity of B1W. As the below figure, it was confirmed that there was no osteoblast activity of B1W. So, we could not proceed the further study for osteoblast activity.

Fig) Osteoblast Activity of B1W. There is no activity at high concentration, 100 μg/mL of B1W.

---

## [Decision Letter · Decision Letter 1]

7 Jan 2025

“Betaone” barley water extract suppresses ovariectomy-induced osteoporosis in vivo and RANKL-induced osteoclast differentiation in vitro

PONE-D-24-41839R1

Dear Dr. Son,

We’re pleased to inform you that your manuscript has been judged scientifically suitable for publication and will be formally accepted for publication once it meets all outstanding technical requirements.

Kind regards,

Liangliang Xu

Academic Editor

PLOS ONE

Additional Editor Comments (optional):

Reviewers' comments:

Reviewer's Responses to Questions

**Comments to the Author**

1. If the authors have adequately addressed your comments raised in a previous round of review and you feel that this manuscript is now acceptable for publication, you may indicate that here to bypass the “Comments to the Author” section, enter your conflict of interest statement in the “Confidential to Editor” section, and submit your "Accept" recommendation.

Reviewer #1: All comments have been addressed

Reviewer #2: All comments have been addressed

Reviewer #3: All comments have been addressed

2. Is the manuscript technically sound, and do the data support the conclusions?

Reviewer #1: Yes

Reviewer #2: Yes

Reviewer #3: Yes

3. Has the statistical analysis been performed appropriately and rigorously? 

Reviewer #1: Yes

Reviewer #2: Yes

Reviewer #3: N/A

4. Have the authors made all data underlying the findings in their manuscript fully available?

Reviewer #1: Yes

Reviewer #2: Yes

Reviewer #3: Yes

5. Is the manuscript presented in an intelligible fashion and written in standard English?

Reviewer #1: Yes

Reviewer #2: Yes

Reviewer #3: Yes

6. Review Comments to the Author

Reviewer #1: The questions have been well answered and can support the conclusions. Articles can be accepted for publication.

Reviewer #2: (No Response)

Reviewer #3: (No Response)

7. PLOS authors have the option to publish the peer review history of their article (what does this mean? ). If published, this will include your full peer review and any attached files.

**Do you want your identity to be public for this peer review?** For information about this choice, including consent withdrawal, please see our Privacy Policy .

Reviewer #1: No

Reviewer #2: No

Reviewer #3: No

---

## [Editor Report · Acceptance letter]

PONE-D-24-41839R1

PLOS ONE

Dear Dr. Son,

I'm pleased to inform you that your manuscript has been deemed suitable for publication in PLOS ONE. Congratulations! Your manuscript is now being handed over to our production team.

Kind regards,

on behalf of

Professor Liangliang Xu

Academic Editor

PLOS ONE